# LincRNA-EPS Promotes Proliferation of Aged Dermal Fibroblast by Inducing CCND1

**DOI:** 10.3390/ijms25147677

**Published:** 2024-07-12

**Authors:** Liping Zhang, Iris C. Wang, Songmei Meng, Junwang Xu

**Affiliations:** 1Department of Physiology, College of Medicine, University of Tennessee Health Science Center, Memphis, TN 38163, USA; lzhan112@uthsc.edu (L.Z.); iris_c_wang@brown.edu (I.C.W.); smeng@uthsc.edu (S.M.); 2Division of Applied Mathematics, Brown University, Providence, RI 02912, USA

**Keywords:** aging, wound healing, dermal fibroblasts, LincRNA-EPS, MiR-34a

## Abstract

The aging process is linked to numerous cellular changes, among which are modifications in the functionality of dermal fibroblasts. These fibroblasts play a crucial role in sustaining the healing of skin wounds. Reduced cell proliferation is a hallmark feature of aged dermal fibroblasts. Long intergenic non-coding RNA (lincRNAs), such as LincRNA-EPS (Erythroid ProSurvival), has been implicated in various cellular processes. However, its role in aged dermal fibroblasts and its impact on the cell cycle and its regulator, Cyclin D1 (CCND1), remains unclear. Primary dermal fibroblasts were isolated from the skin of 17-week-old (young) and 88-week-old (aged) mice. Overexpression of LincRNA-EPS was achieved through plasmid transfection. Cell proliferation was detected using the MTT assay. Real-time PCR was used to quantify relative gene expressions. Our findings indicate a noteworthy decline in the expression of LincRNA-EPS in aged dermal fibroblasts, accompanied by reduced levels of CCND1 and diminished cell proliferation in these aging cells. Significantly, the overexpression of LincRNA-EPS in aged dermal fibroblasts resulted in an upregulation of CCND1 expression and a substantial increase in cell proliferation. Mechanistically, LincRNA-EPS induces CCND1 expression by sequestering miR-34a, which was dysregulated in aged dermal fibroblasts, and directly targeting CCND1. These outcomes underscore the crucial role of LincRNA-EPS in regulating CCND1 and promoting cell proliferation in aged dermal fibroblasts. Our study provides novel insights into the molecular mechanisms underlying age-related changes in dermal fibroblasts and their implications for skin wound healing. The significant reduction in LincRNA-EPS expression in aged dermal fibroblasts and its ability to induce CCND1 expression and enhance cell proliferation highlight its potential as a therapeutic target for addressing age-related skin wound healing.

## 1. Introduction

The elderly population, who currently make up 12.4% (35 million) of the total US population, is rapidly growth and is projected to reach 20% (53 million) by 2030 (U.S. Census, 2000). Among those over the age of 65, chronic wounds such as pressure ulcers, diabetic foot ulcers, and venous leg ulcers are common, affecting about 3% of this age group in the United States [1]. The annual cost of managing chronic wounds in the US is around USD 10 billion, with a significant portion attributed to wound care for adults aged 65 and above [2]. Given the lack of effective treatments for wounds in the elderly, addressing this critical issue is essential.

The aging process is characterized by numerous cellular alterations, including modifications in the functionality of dermal fibroblasts, which play a pivotal role in maintaining skin homeostasis and wound healing [3,4,5,6]. Dermal fibroblasts are key cellular constituents of the dermis and are responsible for synthesizing extracellular matrix components, providing structural support, and orchestrating tissue repair processes in response to injury [3,4,5,6]. However, aging-related changes in dermal fibroblasts significantly impair their ability to effectively participate in wound healing processes.

One hallmark feature of aged dermal fibroblasts is the decline in cell proliferation capacity, which is essential for efficient wound repair [6,7,8]. Compared to young skin (age: 18–29), the total number of fibroblasts in aged skin (over 80 years) is reduced by approximately 35% [9]. Additionally, the number of senescent fibroblasts increases with age, as indicated by a significant rise in p16INK4a-positive cells, a marker of senescence that encodes an inhibitor of CDK4/6, in the dermis of aged human skin [10,11]. This diminished proliferative potential contributes to delayed wound healing observed in elderly individuals. Additionally, aging is associated with alterations in the expression of various regulatory molecules, including non-coding RNAs, which can profoundly impact cellular functions and tissue homeostasis [12,13].

LincRNAs have emerged as important regulators of gene expression and cellular processes, despite lacking protein-coding potential. LincRNAs play diverse roles in gene regulation, chromatin organization, and cellular signaling pathways [14]. However, their involvement in age-related changes in dermal fibroblasts and their implications for wound healing remain poorly understood. Recently, LincRNA-EPS has been identified as a transcriptional regulator that restrains inflammation by inhibiting the expression of immune response genes [15,16,17]. However, its role in aged dermal fibroblasts and its impact on critical regulators of cell cycle progression, such as Cyclin D1 (CCND1), remain largely unexplored. CCND1 is a key cell cycle regulator that promotes G1 to S phase transition and is key for cell proliferation. Dysregulation of CCND1 expression can lead to aberrant cell cycle progression and impaired proliferation, contributing to age-related phenotypes. Another non-coding RNA, miR-34a, is a key regulator of age-related tissue changes and an inducer of cellular senescence [18]. Upregulation of miR-34a induces cell cycle arrest, apoptosis, and senescence by targeting critical cell cycle regulators and proliferation proteins such as CDK4, CDK6, Cyclin D1, Cyclin E2, E2F1, E2F3, and CDC25A [18,19,20,21].

Understanding the molecular mechanisms underlying age-related changes in dermal fibroblasts and identifying potential therapeutic targets to enhance their regenerative capacity are essential for developing strategies to improve wound healing outcomes in the elderly population. Therefore, elucidating the role of LincRNA-EPS in regulating CCND1 expression and cell proliferation in aged dermal fibroblasts holds promise for identifying novel therapeutic avenues to address age-related skin wound healing impairments.

## 2. Results

### 2.1. Reduced LincRNA-EPS and Induced miR-34a Expression in Aged Wounds

We conducted an in vivo wound healing analysis to compare the healing processes in young (17-week-old, n = 5) and aged (88-week-old, n = 5) mice. More details of the wound healing animal study was reported in our recent publication [22]. Our results demonstrated that the dorsal 8 mm excisional wounds in aged mice exhibited a significantly slower rate of closure compared to those in young mice [22]. Further, we examined the expression levels of LincRNA-EPS and miR-34a genes in the wounds of both young and aged mice. miR-34a, known to be a key regulator of cell proliferation, plays a critical role in the aging process. Our findings indicated that wounds in aged mice had lower levels of LincRNA-EPS expression and higher levels of miR-34a expression compared to wounds in young mice (Figure 1A,B) at baseline (D0) and 7 days after injury.

### 2.2. Reduced LincRNA-EPS and Induced miR-34a Expression in Aged Dermal Fibroblasts

Quantitative real-time PCR analysis revealed a significant decline in the expression levels of LincRNA-EPS in aged dermal fibroblasts compared to younger fibroblasts. Specifically, primary dermal fibroblasts were isolated from both young and aged skin and were analyzed at passage 2. LincRNA-EPS expression was approximately 10-fold lower in aged fibroblasts, suggesting a disruption in the regulation of this important regulatory molecule during the aging process (Figure 2A).

Conversely, there was a pronounced increase in the expression of miR-34a, a major aging-related microRNA, in aged fibroblasts. This increase in miR-34a expression was substantial, with 3-fold induction observed in aged fibroblasts compared to younger ones (Figure 2B).

### 2.3. Reduced Cell Proliferation and CCND1 Gene Expression in Aged Dermal Fibroblasts

We compared the cell proliferation rates of young and aged dermal fibroblasts using the MTT assay. The results showed that aged dermal fibroblasts exhibited significantly lower levels of cell proliferation compared to young fibroblasts, as indicated by a reduced optical density at 450 nm (Figure 3A). This decreased proliferation capacity may contribute to impaired wound healing in aged skin. Additionally, we observed a significantly reduced expression level of the CCND1 gene in aged dermal fibroblasts, consistent with their lower proliferation rate (Figure 3B).

### 2.4. Effects of LincRNA-EPS Overexpression on CCND1 Gene Expression and Cell Proliferation

To investigate the functional significance of LincRNA-EPS in aged dermal fibroblasts, we employed a plasmid-based approach to overexpress LincRNA-EPS in aged fibroblasts. Remarkably, the overexpression of LincRNA-EPS (pEPS) resulted in a notable upregulation of CCND1 expression in aged dermal fibroblasts, indicating a regulatory role for LincRNA-EPS in modulating CCND1 levels (Figure 4A). Assessment of cell proliferation using the MTT assay demonstrated a significant increase in proliferation rates in aged dermal fibroblasts following LincRNA-EPS overexpression (Figure 4B). Cell proliferation maker gene MKI67 was significantly induced in LincRNA-EPS overexpressing aged dermal fibroblasts (Figure 4C).

### 2.5. Effects of miR-34a Overexpression on CCND1 Gene Expression and Cell Proliferation

In aged dermal fibroblasts, the decreased cell proliferation was associated with a reduced expression level of CCND1. Given that miR-34a is a crucial regulator of the cell cycle and that CCND1 is a confirmed target of miR-34a (Figure 5A), we investigated the impact of miR-34a overexpression on cell proliferation. Overexpression of miR-34a significantly decreased CCND1 gene expression in both young and aged dermal fibroblasts (Figure 5B). This reduction in CCND1 expression was accompanied by a significant decrease in cell proliferation rates, as measured by the MTT assay, particularly in aged dermal fibroblasts overexpressing miR-34a (Figure 5C).

### 2.6. Possible Mechanisms of LinRNA-EPS/miR-34a Signaling in Aged Dermal Fibroblast

The effects of LincRNA-EPS and miR-34a on cell proliferation are opposing. LincRNA-EPS overexpression promotes cell proliferation, whereas miR-34a overexpression inhibits it, both by regulating CCND1 gene expression. To understand the mechanism by which LincRNA-EPS regulates CCND1 expression, we analyzed miR-34a levels in cells overexpressing LincRNA-EPS. Our findings revealed that miR-34a expression was significantly reduced in these cells, suggesting that LincRNA-EPS acts as a molecular sponge to sequester miR-34a (Figure 6B). Bioinformatics analysis identified a miR-34a binding site on LincRNA-EPS (Figure 6A), and real-time PCR confirmed that LincRNA-EPS levels decrease in miR-34a-overexpressing cells (Figure 6C). This indicates that LincRNA-EPS regulates cell proliferation by modulating miR-34a availability and thereby influencing CCND1 gene expression (Figure 6D).

## 3. Discussion

Our study provides compelling evidence for the involvement of LincRNA-EPS in age-related alterations in dermal fibroblasts. The significant decrease in LincRNA-EPS expression observed in wounds of aged skin and aged fibroblasts suggests its potential contribution to the impaired proliferative capacity of these cells, which is a hallmark feature of aging-related wound healing deficiencies.

LincRNA-EPS was initially reported to inhibit apoptosis during erythroid differentiation in part through repressing the expression of the proapoptotic gene Pycard [17]. Recently, LincRNA-EPS, identified as a novel long intergenic RNA, emerges as a pivotal regulator in the realm of inflammation. Initially recognized for its role in restraining the inflammatory response of macrophages, mitigating inflammation in vital organs, and reducing lethality following endotoxin challenge [15], subsequent studies have unveiled its anti-inflammatory effects in diverse contexts such as acute pancreatitis [16], cerebral infarction [23], and active pulmonary tuberculosis [24]. In this study, we discovered a novel function of LincRNA-EPS that it can promote cell proliferation.

Our findings elucidate a novel mechanism by which LincRNA-EPS regulates CCND1 expression in aged dermal fibroblasts. LincRNA-EPS acts as a molecular sponge for miR-34a, a microRNA dysregulated in aging, thereby alleviating its inhibitory effect on CCND1 expression. This regulatory role of LincRNA-EPS in modulating CCND1 expression underscores its significance in regulating cell cycle progression and proliferation in aged dermal fibroblasts.

The restoration of CCND1 expression and cell proliferation observed upon LincRNA-EPS overexpression in aged dermal fibroblasts holds promising therapeutic implications for age-related skin wound healing. Given the critical role of dermal fibroblasts in orchestrating wound repair processes, enhancing their proliferative capacity through targeted interventions such as LincRNA-EPS modulation could potentially mitigate the delayed wound healing observed in elderly individuals.

Targeted interventions aimed at restoring LincRNA-EPS expression or modulating its regulatory interactions may hold promise for improving wound healing outcomes in aging populations. However, comprehensive preclinical studies are warranted to assess the efficacy, specificity, and potential off-target effects of such interventions before clinical translation.

While our study provides valuable insights into the molecular mechanisms underlying age-related changes in dermal fibroblasts, several avenues for future research merit exploration. Further elucidation of the signaling pathways and molecular networks involved in LincRNA-EPS-mediated regulation of CCND1 expression and cell proliferation could provide deeper insights into its therapeutic potential.

In conclusion, our study sheds light on the critical role of LincRNA-EPS in regulating CCND1 expression and cell proliferation in aged dermal fibroblasts. The identification of LincRNA-EPS as a key regulator of age-related changes in dermal fibroblast function underscores its potential as a therapeutic target for addressing age-related skin wound healing impairments. Further research aimed at harnessing the therapeutic potential of LincRNA-EPS modulation holds promise for improving wound healing outcomes in the elderly population.

## 4. Materials and Methods

### 4.1. Animal Studies

All animal experiments received approval from the Institutional Animal Care and Use Committee at the University of Tennessee Health Science Center and adhered to the NIH Guide for the Care and Use of Laboratory Animals. We used female C57BL/6J mice aged 17 weeks (young) and 88 weeks (aged) from the Jackson Laboratory (Bar Harbor, ME, USA). The mice were anesthetized with inhaled isoflurane. Their dorsal skin was shaved, depilated, and swabbed with alcohol and Betadine (Purdue Pharma, Stamford, CT, USA) before wounding. Each mouse was given a single full-thickness dorsal wound, including the panniculus carnosum, using an 8 mm punch biopsy (Miltex Inc., York, PA, USA). The wounds were dressed with Tegaderm (3M, St Paul, MN, USA), which was removed on postoperative day 2. Postoperatively, the mice received a subcutaneous injection of the analgesic carprofen. Wound images were taken every other day and analyzed using ImageJ (https://imagej.net/software/imagej/). Full-thickness skin samples centered on the wound were collected at 0 and 7 days after surgery (n = 5 per timepoint).

### 4.2. Dermal Fibroblast Isolation and Culture

Primary dermal fibroblasts were isolated from the skin wounds of 17-week-old (young) and 88-week-old (aged) mice. The cells were maintained at 37 °C in a humidified atmosphere with 5% CO2 and cultured in Dulbecco’s Modified Eagle Medium high-glucose (DMEM, Sigma-Aldrich, St. Louis, MO, USA) containing 10% fetal bovine serum (FBS; Gibco, MA, USA) and 1% antibiotic–antimycotic (Sigma-Aldrich, St. Louis, MO, USA). Cells at passage 2 were used for in vitro experiments.

### 4.3. Cell Transfection

The LincRNA-EPS overexpression plasmid (pEPS) and the control plasmid pcDNA-3.1 (p3.1) were obtained from Applied Biological Materials Inc. (Richmond, BC, Canada). These expression vectors were transfected into fibroblasts using the liposome transfection reagent Lipofectamine 2000 (Invitrogen). The efficiency of overexpression was confirmed via reverse transcription quantitative polymerase chain reaction (RT-qPCR). For cell mimic transfection, fibroblasts were seeded into a 6-well plate and transfected with either miR-34a mimic or miR-NC (50 nM, Life Technologies, Carlsbad, CA, USA), also using Lipofectamine 2000. Cells were harvested 24 h post-transfection, and the overexpression efficiency was verified through RT-qPCR.

### 4.4. Real Time Quantitative PCR

Total RNA was extracted using TRIzol reagent (Invitrogen, Carlsbad, CA, USA) following the manufacturer’s protocol. The RNA was then converted into cDNA with the SuperScript First-Strand Synthesis System (Invitrogen, Life Technologies). For gene expression analysis, LincRNA-EPS, CCND1, and MKI67 were amplified using the TaqMan gene expression assay (Applied Biosystems, Waltham, MA, USA) with GAPDH as the internal normalization control. Quantitative RT-PCR analyses for miR-34a and RNU6 (normalization control) were conducted using TaqMan miRNA assays with reagents, primers, and probes from Qiagen. Samples (n = 5 per group) were amplified in triplicate, and the results for each sample were averaged. Relative gene expression was calculated using the ΔΔCT method and reported as mean ± SD.

### 4.5. Cell Proliferation MTT Assay

To perform the MTT assay, fibroblast cells were seeded at a density of 2000 cells per well in 96-well plates. After incubation for 2 days, 20 µL of MTT reagent (5 mg/mL, cat. no. ab211091; Abcam, Cambridge, MA, USA) was added to each well. The plates were then incubated at 37 °C for 1 h to allow the formation of formazan crystals. Following incubation, the remaining formazan crystals were dissolved by adding dimethyl sulfoxide (DMSO) to each well. The absorbance of the resulting solution was measured at 450 nm using a microplate reader to determine cell viability.

### 4.6. Statistical Analysis

Results are shown as mean ± SD based on 3 to 5 independent experiments. Differences in gene expression between the two groups were analyzed using the Student’s *t*-test, with a *p*-value of less than 0.05 indicating statistical significance. 

## Figures and Tables

**Figure 1 ijms-25-07677-f001:**
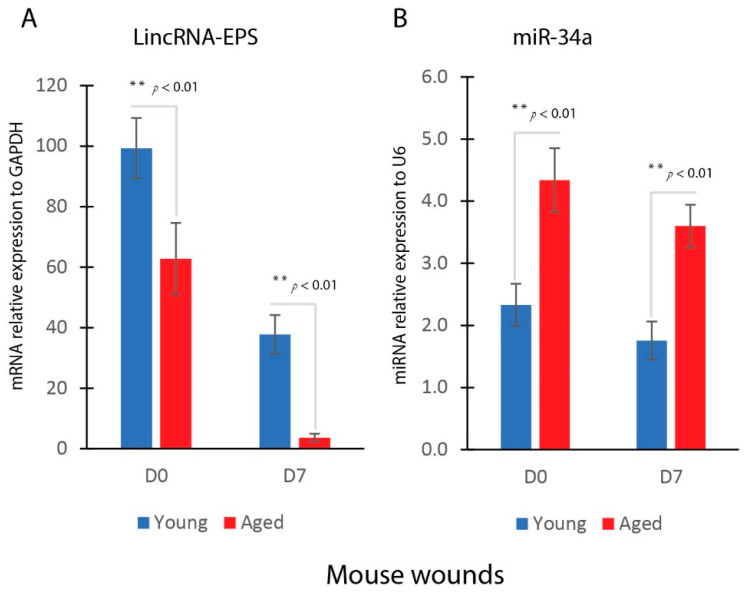
Reduced LincRNA-EPS and induced miR-34a expression in aged wounds. (**A**) Real-time qPCR analysis of LincRNA-EPS gene expression in young and aged mice wounds (mean ± SD, n = 3 per group). (**B**) Real-time qPCR analysis of miR-34a gene expression in young and aged mice wounds (mean ± SD, n = 3 per group). ** *p* < 0.01.

**Figure 2 ijms-25-07677-f002:**
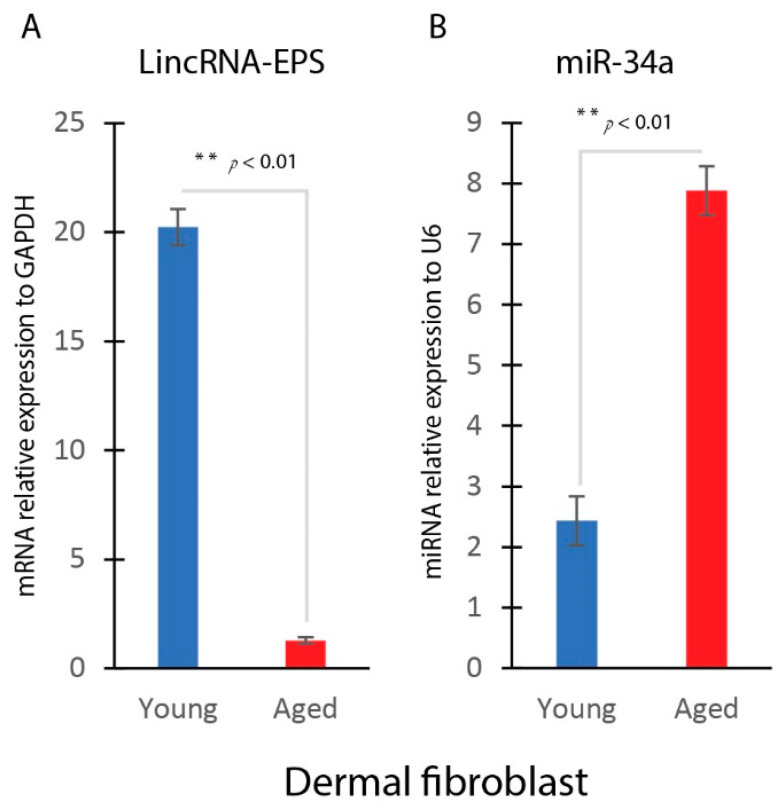
Reduced LincRNA-EPS and induced miR-34a expression in aged dermal fibroblasts. (**A**) Real-time qPCR analysis of LincRNA-EPS gene expression in young and aged dermal fibroblast (mean ± SD, n = 3 per group). (**B**) Real-time qPCR analysis of miR-34a gene expression in young and aged dermal fibroblast (mean ± SD, n = 3 per group). ** *p* < 0.01.

**Figure 3 ijms-25-07677-f003:**
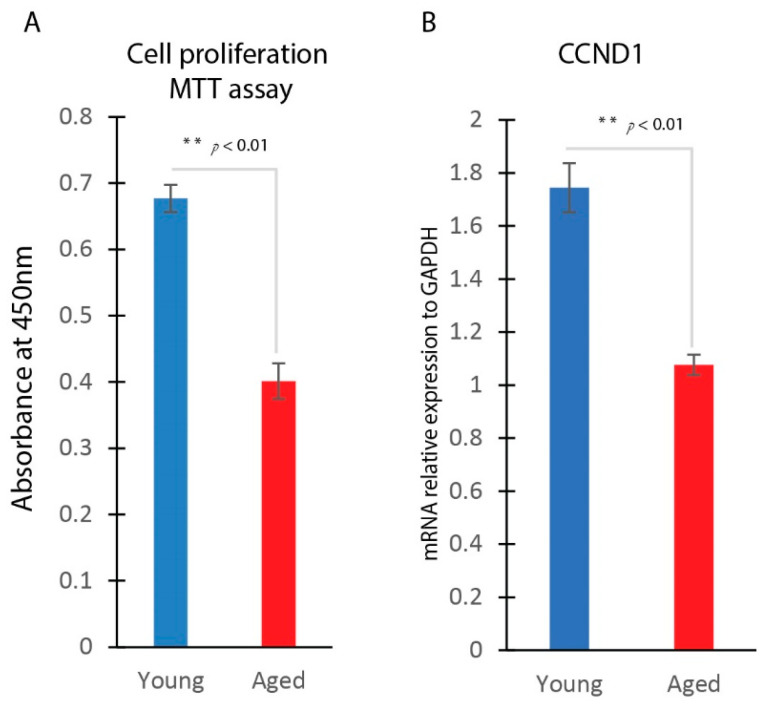
Reduced cell proliferation and CCND1 gene expression in aged dermal fibroblasts. (**A**) Young and aged dermal fibroblast cell proliferation analysis using MTT assay. (**B**) Real-time qPCR analysis of CCND1 gene expression in young and aged dermal fibroblast (mean ± SD, n = 3 per group). ** *p* < 0.01.

**Figure 4 ijms-25-07677-f004:**
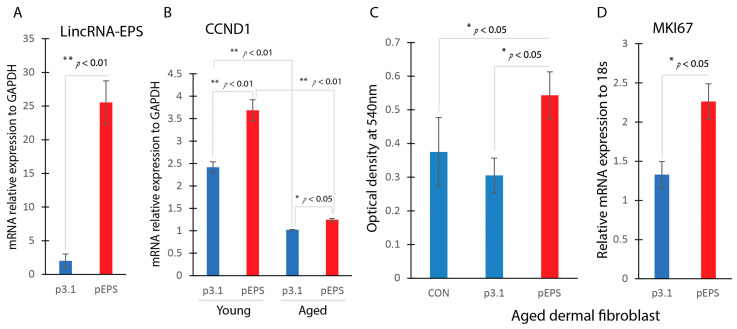
LincRNA-EPS overexpression induces CCND1 and cell proliferation. Dermal fibroblasts were transfected with pEPS (plasmid to overexpress LincRNA-EPS) or control plasmid (p3.1). (**A**) LincRNA-EPS overexpression via plasmid transfection confirmed by RT-qPCR. (**B**) The gene expression level of CCND1 was determined by RT-qPCR in EPS overexpression fibroblast or control transfected fibroblasts. (**C**) Cell proliferation between aged dermal fibroblast transfected with pEPS or control p3.1 using MTT assay. (**D**) Cell proliferation maker gene MKI67 was determined by RT-qPCR.

**Figure 5 ijms-25-07677-f005:**
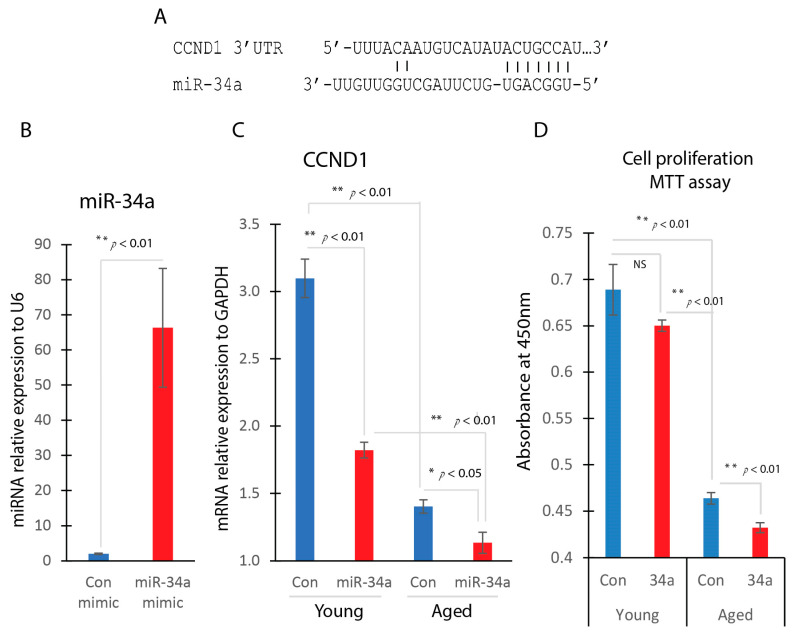
MiR-34a overexpression reduces CCND1 and cell proliferation. (**A**) Bioinformatic analysis identified miR-34a binding site on CCND1 3′ UTR. (**B**) TR-qPCR analysis indicated that miR-34a was significantly induced in miR-34a mimic transfected cells. (**C**) Dermal fibroblasts were transfected with miR-34a mimic (miR-34a) or control mimic (Con). The gene expression level of CCND1 was determined by RT-qPCR in miR-34a overexpression fibroblast or control transfected fibroblasts. (**D**) Cell proliferation between dermal fibroblast transfected with miR-34a or control using MTT assay.

**Figure 6 ijms-25-07677-f006:**
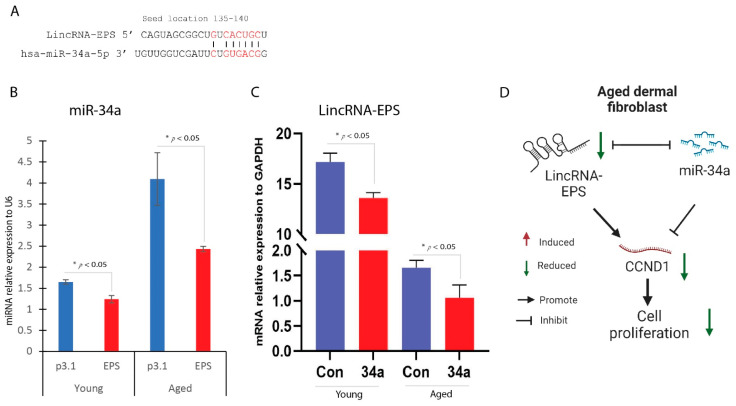
miR-34a and LincRNA-EPS regulate each other. (**A**) miR-34a binding site on LincRNA-EPS by bioinformatics analysis. (**B**) Real-time qPCR analysis of miR-34a gene expression in young and aged dermal fibroblasts with EPS overexpression (mean ± SD, n = 3 per group). (**C**) Real-time qPCR analysis of LincRNA-EPS gene expression in young and aged dermal fibroblasts with miR-34a overexpression (mean ± SD, n = 3 per group). * *p* < 0.05. (**D**) Possible mechanisms of LinRNA-EPS/miR-34a signaling in aged dermal fibroblast.

## Data Availability

Data are contained within the article.

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
