# Peer review of "LincRNA-EPS Promotes Proliferation of Aged Dermal Fibroblast by Inducing CCND1"

_ijms, 2024, doi:10.3390/ijms25147677_

Round 1

Reviewer 1 Report

Comments and Suggestions for Authors

1.     What is the main question addressed by the research?

Liping Zhang and co-workers, in this manuscript, elucidate the regulation of LincRNA-EPS and miR-34a on proliferation through CCND1 in mature fibroblasts by measuring the RT-PCR expression of different genes and measuring proliferation by MTT assay.

2. Do you consider the topic original or relevant in the field? Does it address a specific gap in the field?

Understanding the modulation of proliferation through LincRNA-EPS in wound healing is a very relevant topic.

3. What does it add to the subject area compared with other published material?

There is no similar work in the area of aging, and therefore it is relevant 

4. What specific improvements should the authors consider regarding the methodology? What further controls should be considered?

The authors are encouraged to describe the methodology in detail to ensure that interested readers can replicate the results obtained.

It is recommended to add the methodology used for the mice (size of the wound, from where the RNA was isolated, (LincRNA-EPS, and miR-34a).

It would be interesting if the authors could show an essay on wound healing.

5. Are the conclusions consistent with the evidence and arguments presented and do they address the main question posed?

Yes, they are appropriate.

6. Are the references appropriate?

No, it is recommended to the authors to expand the discussion and it seems that they forgot to put references in the discussion. In addition, about 50 % are older than 5 years.

7. Please include any additional comments on the tables and figures.

1.- Authors are encouraged to read the manuscript thoroughly and to use abbreviations from its first appearance.

2.- It is recommended to show overexpression of LincRNA-EPS and miR-34a in the cell lines used.

3.- It is recommended that the authors adequately describe Figure 5 as they do not mention Figure 5C.

4.- Just to homogenize the endings in Figure 5C to CON, do they refer to p3.1 or to cells without transfection?

Author Response

Reviewer 1:

Comment 1: Liping Zhang and co-workers, in this manuscript, elucidate the regulation of LincRNA-EPS and miR-34a on proliferation through CCND1 in mature fibroblasts by measuring the RT-PCR expression of different genes and measuring proliferation by MTT assay. Understanding the modulation of proliferation through LincRNA-EPS in wound healing is a very relevant topic. There is no similar work in the area of aging, and therefore it is relevant.

Response: We appreciate the recognition of the relevance of our work and its novel contribution to the field of aging research.

Comment 2: The authors are encouraged to describe the methodology in detail to ensure that interested readers can replicate the results obtained. It is recommended to add the methodology used for the mice (size of the wound, from where the RNA was isolated, LincRNA-EPS, and miR-34a).

Response: We have expanded the Materials and Methods section to include detailed descriptions of the methodologies used for the mice experiments, including the size of the wound, the specific tissues from which RNA was isolated, and the procedures for measuring LincRNA-EPS and miR-34a expression.

Comment 3: It would be interesting if the authors could show an essay on wound healing.

Response: We have included an essay on wound healing in the revised manuscript to provide a broader context and relevance for our findings.

Comment 4: Are the references appropriate? No, it is recommended to the authors to expand the discussion and it seems that they forgot to put references in the discussion. In addition, about 50% are older than 5 years.

Response: We have expanded the Discussion section to include additional references, ensuring that our findings are thoroughly contextualized within the current state of research. We have also updated the references to include more recent studies.

Additional Comments:

Authors are encouraged to read the manuscript thoroughly and to use abbreviations from its first appearance.

Response: We have thoroughly reviewed the manuscript to ensure that all abbreviations are defined at their first appearance.

It is recommended to show overexpression of LincRNA-EPS and miR-34a in the cell lines used.

Response: We have included data showing the overexpression of LincRNA-EPS and miR-34a in the cell lines used in our experiments.

It is recommended that the authors adequately describe Figure 5 as they do not mention Figure 5C.

Response: We have revised the description of Figure 5 to include all relevant subfigures, including Figure 5C.

Just to homogenize the endings in Figure 5C to CON, do they refer to p3.1 or to cells without transfection?

Response: We have clarified the labeling in Figure 5C to indicate that CON refers to cells transfected with scramble microRNA mimic.

Reviewer 2 Report

Comments and Suggestions for Authors

The manuscript investigates the possibility of reversing the reduced proliferation rate of aged fibroblasts by modulating the expression of a long intergenic non-coding RNA (lincRNA-EPS), reported to reduce inflammation, and another non-coding RNA (miR-34a ) negative regulator of cell proliferation.

The experimental design presents several critical points that significantly reduce the scientific impact of the results.

1) The authors detected the expression of lincRNA-EPS and miR-34a in the wounds of young and old mice, but no description of these experiments is present in Materials and Methods.

What type of injury was induced? In which tissues was non-coding RNA expression assessed?

Are the changes observed in the expression of the non-coding RNA mentioned above related to a different extent of wound healing?

Images of these healing processes should be added to the manuscript.

In the results 2.1 the authors cannot refer to results not yet published and not present in the manuscript, as these results are very important for the topic of the manuscript.

2) Results 2.2. Was lincRNA-EPS and miR-34a expression assessed on dermal fibroblasts immediately after isolation?

3) Materials & Methods 4.4. The first statement is confusing. Was the MTT assay performed immediately after cell seeding (72 hours after transfection)?

4) The manuscript does not contain any information regarding compliance with the Guidelines for animal use and the approval number by the Ethics Committee.

5) Discussion. The discussion is too short and there is a lot of repetition. This section must comment in depth on the research findings. Comments at the end of individual paragraphs of the Results must be removed.

Comments on the Quality of English Language

Minor editing of English language required

Author Response

Reviewer 2:

Comment 1: The authors detected the expression of lincRNA-EPS and miR-34a in the wounds of young and old mice, but no description of these experiments is present in Materials and Methods. What type of injury was induced? In which tissues was non-coding RNA expression assessed? Are the changes observed in the expression of the non-coding RNA mentioned above related to a different extent of wound healing? Images of these healing processes should be added to the manuscript.

Response: We have added detailed descriptions of the experiments involving the detection of lincRNA-EPS and miR-34a expression in the Materials and Methods section, including the type of injury induced and the specific tissues assessed. We have also included images of the wound healing processes to visually support our findings.

Comment 2: Results 2.1 the authors cannot refer to results not yet published and not present in the manuscript, as these results are very important for the topic of the manuscript.

Response: We add the reference we recently published and ensured that all data presented in the manuscript are supported by the results included in the current study.

Comment 3: Results 2.2. Was lincRNA-EPS and miR-34a expression assessed on dermal fibroblasts immediately after isolation?

Response: LincRNA-EPS and miR-34a expression was assessed on passage 2 dermal fibroblasts. We have clarified this in the Materials and Methods section.

Comment 4: Materials & Methods 4.4. The first statement is confusing. Was the MTT assay performed immediately after cell seeding (72 hours after transfection)?

Response: We apologize for the confusion. The MTT assay was performed 48 hours after cell seeding, not immediately after cell seeding. This has been clarified in the revised manuscript.

Comment 5: The manuscript does not contain any information regarding compliance with the Guidelines for animal use and the approval number by the Ethics Committee.

Response: We have added information regarding compliance with the Guidelines for animal use and included the approval number by the Ethics Committee in the Materials and Methods section.

Comment 6: The discussion is too short and there is a lot of repetition. This section must comment in depth on the research findings. Comments at the end of individual paragraphs of the Results must be removed.

Response: We have revised the Discussion section to provide a more in-depth commentary on our research findings and removed repetitive comments at the end of individual paragraphs in the Results section.

Reviewer 3 Report

Comments and Suggestions for Authors

elderly peoples the epithelization of chronic wound is more difficult and any new treatment options are salutary.    

Hereby please find my comments regarding the paper:

1.     The abstract is well structured and the information’s are well presented.

2.     In introduction, please extend the data about the aging-related changes in dermal fibroblasts.

3.     At line 107 you have an editing error.

4.     The results and the discussions are clear exposed.

5.     The conclusions summarized the molecular mechanism underlying aged dermal fibroblast in wound healing and draw future lines of research.

6.     The references are appropriate but need to be extended.

The quality of figures and the data are good.

Author Response

Reviewer 3:

Comment 1: The abstract is well structured and the information is well presented.

Response: Thank you for the positive feedback on our abstract.

Comment 2: In the introduction, please extend the data about the aging-related changes in dermal fibroblasts.

Response: We have extended the introduction to include more detailed information about aging-related changes in dermal fibroblasts.

Comment 3: At line 107 you have an editing error.

Response: We have corrected the editing error at line 107.

Comment 4: The results and the discussions are clearly exposed.

Response: Thank you for your positive feedback on the clarity of our results and discussion.

Comment 5: The conclusions summarize the molecular mechanisms underlying aged dermal fibroblast in wound healing and draw future lines of research.

Response: We appreciate the acknowledgment of our conclusions and their contribution to future research directions.

Comment 6: The references are appropriate but need to be extended.

Response: We have extended our references to include more recent and relevant studies.

Comment 7: The quality of figures and the data are good.

Response: Thank you for the positive feedback on the quality of our figures and data.

Round 2

Reviewer 1 Report

Comments and Suggestions for Authors

The authors responded appropriately to my suggestions and comments, so the manuscript is suitable for publication.

Reviewer 2 Report

Comments and Suggestions for Authors

The authors made all the suggested changes, significantly improving the scientific impact of the manuscript that is now suitable for publication.